# Advances in Anti-Diabetic Cognitive Dysfunction Effect of Erigeron Breviscapus (Vaniot) Hand-Mazz

**DOI:** 10.3390/ph16010050

**Published:** 2022-12-29

**Authors:** Shanye Gu, Ziyi Zhou, Shijie Zhang, Yefeng Cai

**Affiliations:** 1The Second Clinical College, Guangzhou University of Chinese Medicine, Guangzhou 510006, China; 2Department of Neurology, Guangdong Provincial Hospital of Chinese Medicine, Guangzhou 510120, China; 3Guangdong Provincial Key Laboratory of Research on Emergency in Traditional Chinese Medicine, Guangzhou 510120, China

**Keywords:** diabetic cognitive dysfunction, pathogenesis, Erigeron breviscapus (vant.) hand-mazz., active ingredients, pharmacological effects

## Abstract

Diabetic cognitive dysfunction (DCD) is the decline in memory, learning, and executive function caused by diabetes. Although its pathogenesis is unclear, molecular biologists have proposed various hypotheses, including insulin resistance, amyloid β hypothesis, tau protein hyperphosphorylation hypothesis, oxidative stress and neuroinflammation. DCD patients have no particular treatment options and current pharmacological regimens are suboptimal. In recent years, Chinese medicine research has shown that herbs with multi-component, multi-pathway and multi-target synergistic activities can prevent and treat DCD. Yunnan is home to the medicinal herb Erigeron breviscapus (Vant.) Hand-Mazz. (EBHM). Studies have shown that EBHM and its active components have a wide range of pharmacological effects and applications in cognitive disorders. EBHM’s anti-DCD properties have been seldom reviewed. Through a literature study, we were able to evaluate the likely pathophysiology of DCD, prescribe anti-DCD medication and better grasp EBHM’s therapeutic potential. EBHM’s pharmacological mechanism and active components for DCD treatment were also summarized.

## 1. Introduction

Diabetic cognitive dysfunction (DCD) is a neurophysiological and anatomical brain modification induced by diabetes mellitus (DM). It causes memory loss, learning capacity decrease, language impairment, and mental and behavioral abnormalities. Monte and Wands [1] described it in 1922. Based on illness course or severity, cognitive function abnormalities are characterized as asymptomatic preclinical, mild cognitive impairment (MCI) and dementia. Vascular dementia (VaD) and Alzheimer’s disease (AD) are the primary types of diabetes-induced dementia [2].

An elderly generation, changing food trends, and a faster pace of living have increased the prevalence of DM. A recent epidemiological survey showed that type 1 diabetes increased during the COVID-19 pandemic [3].Diabetes causes acute and chronic problems and impairs cognitive function [4]. 1.5–2.0 times more likely to suffer cognitive impairment, or dementia than individuals without diabetes [2,5,6]. Diabetes increases the incidence of MCI and AD by 1.25 to 1.91 times, according to 144 prospective studies [7]. Mild cognitive impairment is common in individuals with type 2 diabetes globally, especially in China and Asia, according to recent research [8].

Although the probable mechanism of DCD is yet unknown, its occurrence is recognized to include a complicated regulatory network. Hypoglycemia, hyperglycemia, insulin resistance, oxidative stress and neuroinflammation, cerebral microvascular injury, gut dysbacteriosis, and amyloid β-peptide (Aβ) buildup are probable pathophysiological processes. Potential treatments for DCD include antidiabetic drugs, weight loss, and cognition-related drugs. For example, ChEIs and memantine, which are commonly used to improve cognition, actually have no significant effect on DCD [9,10].

Traditional Chinese Medicine (TCM), a significant component of world medicine, is essential in the treatment of DCD. The possibility of TCM in the treatment of DCD has been the subject of extensive research and theoretical support from an increasing number of academics in recent years [11,12]. For example, the cinnamon aqueous extract may improve learning ability by enhancing insulin sensitivity and inhibiting cholinesterase activity in a non-transgenic Alzheimer’s disease rat model [13]; Ganoderma (Lingzhi) may help restore the memory and learning ability of an AD animal model by maintaining the structure and function of neurons [14]; Laminaria japonica can significantly reduce the oxidative stress level of cognitive deficit mice induced by amyloid β peptide (Aβ), reduce the expression of Tau protein and neuronal apoptosis [15]. In Yunnan Province, at altitudes ranging from 1700 to 3000 m, Erigeron breviscapus (Vant.) Hand-Mazz. (EBHM) is an annual herb plant of the genus Euphorbia in the Asteraceae family. It thrives in grassy and open woodlands on sunny slopes. The pharmacological benefits of EBHM and its active components, such as baicalin and scutellarin, include increased blood flow, an anti-inflammatory reaction, an anti-oxidative stress response, as well as inhibition of apoptosis [16].

However, the anti-DCD mechanism of EBHM and its active components remains unknown. We outline the active components and pharmacological effects of EBHM and explore the pharmacological mechanisms of EBHM for the treatment of DCD by evaluating the findings of studies on the mechanisms of EBHM for the treatment of DCD. This suggests novel therapy options for DCD and may encourage the clinical use of EBHM.

## 2. Active Ingredients and Pharmacological Effects of EBHM

The Southern Yunnan Materia Medica is where EBHM, often referred to as Dengzhan Asarum, originally appeared. It is categorized as fleabane in the Asteraceae family and is a member of the short pavilion fleabane group. There are about 200 species of erigeron breviscapus that blossom all year, but only three that can be utilized as medication. Whole grass is traditionally harvested in the summer and fall for use as medicine. Along with being often utilized in therapeutic settings, it can also be used daily to make dishes, medicinal liquor and so on. Clinically, EBHM and its preparations are mainly used for the treatment of cardiovascular and cerebrovascular diseases and various types of inflammation. In recent years, studies have found that it has a good therapeutic effect on diabetes, kidney disease, glaucoma, liver damage and other diseases [17].

EBHM contains flavonoids and caffeoylquinic acid (CQA) among its chemical constituents. Due to their medicinal benefits, flavonoids and CQAs are regarded as the key active components in EBHM. Flavonoids and glycosides, such as scutellarin, baicalin and quercetin, have been identified from EBHM (accounting for roughly 47.41% of all known EBHM compounds). Caffeic acid, 5-O-Caffeoylquinic acid and 3,5-Dicaffeoylquinic acid, collectively known as CQA, make up 29.7% of all EBHM substances that have been identified.

The bulk of the chemical components of EBHM are concentrated in the ethyl acetate portion of the ethanol extract and comprise substances with these qualities, such as 3-hydroxy-baicalin, 3-hydroxy-7-methoxy baicalin, 5,7,4’-tri-hydro flavone, cinnamic acid, methyl caffeate, and others. However, gut microorganisms convert the majority of flavonoids. Additionally, phenolic acids are extensively metabolized in the body and produce a variety of metabolites. Numerous metabolites are still undiscovered due to the limits of experimental settings and analytical techniques. Therefore, further research on EBHM metabolites is necessary [18].

It has been shown that the key EBHM ingredients, including scutellarin, baicalin, quercetin and CQA, have favorable pharmacological effects and therapeutic potential for DCD (Figure 1).

The major ingredient of commercially accessible preparations of EBHM is scutellarin, which is the glycosyloxyflavone with a molecular weight of 462.36 [17,19]. Scutellarin has been demonstrated to ameliorate cognitive impairment, anti-amyloid β-peptide (Aβ) toxicity, energy metabolism, suppress oxidative stress and neuroinflammation, stimulate astrocytes, increase the proliferation and differentiation of brain cells, and prevent apoptosis [20].

Another active flavonoid derived from EBHM, baicalin, has been found to have potent pharmacological actions, including the prevention of neural cell damage and apoptosis as well as the scavenging of oxide free radicals and antioxidants [21].

Quercetin is a phytogenic flavonoid present in plants such as fruits, vegetables, and herbs and is one of the flavonoids in EBHM [17]. Its molecular formula is C_21_H_18_O_13_, and it has been claimed to have antioxidant and free radical scavenging, anticancer, anti-inflammatory, expectorant, cough suppressant, asthma soothing, hypotensive, hypolipidemic, and hypoglycemic characteristics. Quercetin may alleviate behavioral dysfunction and cognitive impairment in the development of neurodegenerative disorders [22].

5-O-Caffeoylquinic acid (5-CQA) and 3,5-Dicaffeoylquinic acid (3, 5-di-CQA) are the two main CQAs that may reduce the symptoms of DCD. However, current studies are only focused on the effects of modifying α-glucosidase [23] and minimizing the development of hippocampal Aβ plaques and neuronal loss [24].

## 3. Anti-DCD Effects of EBHM

Previous research has established the precise mechanism of EBHM and its active components in the treatment of Alzheimer’s disease. Dong and Qu demonstrated that EBHM and its active substances (scutellarin, baicalin, and CQA) improved learning memory in animal models of Alzheimer’s disease, and the mechanism may be related to inhibition of Aβ aggregation, modulation of the cholinergic nervous system, reduction of tau protein hyperphosphorylation, resistance to oxidative stress and inflammation, and apoptosis resistance [18]. However, research on the specific mechanism of EBHM and its active components in the treatment of DCD is lacking. Insulin resistance, oxidative stress, neuroinflammation, endoplasmic reticulum (ER) stress, blood-brain barrier damage, apoptosis, gut dysbacteriosis, and aggregation of β-amyloid and tau proteins are now being studied as underlying pathophysiological processes of DCD [10,25,26].

As a result, we went into detail about the anti-DCD effects of EBHM and its active ingredients from the various aspects of DCD pathogenesis (Table 1) and thoroughly summarized and analyzed the pharmacological pathways of action of the active ingredients of EBHM in the therapies of DCD, offering hints and a foundation for further research on new approaches to treating DCD.

### 3.1. Targeted Insulin Resistance

One of the main characteristics of T2DM and a potential risk for dementia is insulin resistance (IR). In animal models of diabetes and fixed effect model of T2DM and AD, impaired insulin signaling pathways are crucial for the deterioration in learning and memory, and T2DM may influence the pathophysiology of AD [51,52].

The brain has many insulin receptors, with the olfactory bulb, hypothalamus, hippocampus, cerebral cortex, striatum, and cerebellum having the highest densities. On the one hand, brain insulin resistance reduces the serine phosphorylation of insulin receptor substrate 1, protein kinase B phosphorylation through cascade reactions, and diminished phosphorylation of c-Jun amino-terminal kinase (JNK), which in turn impacts insulin pathway components such as glycogen synthase kinase-3 and protein kinase A. The insulin signaling pathway may be activated by changes in the activity of certain downstream components, which can promote tau protein hyperphosphorylation and neurofibrillary tangle development. In addition, by raising insulin levels, these downstream components can compete with Aβ for insulin-degrading enzymes, affecting Aβ clearance [53]. The modified activity of these downstream components can also promote tau protein hyperphosphorylation and the formation of neurofibrillary tangles, which can impair insulin signaling. This shows that inhibiting Aβ deposition and tau protein hyperphosphorylation requires lowering insulin resistance.

The beta subunit of the tetrameric insulin receptor, which contains kinase activity and phosphorylates the insulin receptor substrates (IRS), is one of two forms of the receptor. Several proteins use IRS as a mooring point, and it may bind to or activate proteins containing an SH2 domain, such as phosphatidylinositol 3-kinase (PI3K). Insulin primarily controls the PI3K pathway to control metabolism. Glucose transporter-4 (GLUT-4) is promoted to move to the cell membrane by protein kinase B (PKB, also known as AKT), a signaling molecule downstream of PI3K for glucose absorption [54,55].

AMP-activated protein kinase (AMPK), which is present in DM, controls IR and glucose metabolism. AMPK and glucose metabolism are closely related. It can be activated by the downstream enzymes glucose-6-phosphatase (G-6-Pase) and phosphoenolpyruvate carboxykinase (PEPCK), which affects glucose production and improves DM. It can also be activated by the upstream enzymes liver kinase B1 (LKB1), transforming growth factor β-activated kinase 1 (TAK1), and Ca2+/Calmodulin-dependent protein kinase kinase β (CaMKKβ). Additionally, GLUT4 can be controlled by AMPK to enhance IR. Low glucose, hypoxia, ischemia, and heat shock typically activate an AMPK signaling pathway in vivo [56], which then encourages ATP synthesis and restricts ATP consumption in various tissues. The transcription and translocation of GLUT4 are enhanced by AMPK activation, increasing the uptake of glucose-induced by insulin [57]. By inhibiting ACC and activating PFK2, it also promotes catabolic processes like fatty acid oxidation and glycolysis [58].

Scutellarin (50 mg/kg) could significantly improve islet function, lower body weight, lower fasting blood glucose and fat index levels and promote glucose uptake in the C57BL/6 mice model induced by a high-fat diet (HFD) after 16 weeks of administration. This suggests that scutellarin could activate the AMPKα-mediated insulin signaling pathway by up-regulating p85α, activating the PI3K/AKT pathway and ultimately affecting the expression of GLUT4 (Figure 2). Additionally, SOD levels may be raised by scutellarin, which likewise has an anti-oxidative stress action [30].

Baicalin may decrease adenylate-activated protein kinase levels, increase glucose absorption and glycolysis, and block hepatocyte gluconeogenesis via activating the AMPK-mediated PI3K/AKT pathway [59], which reduces adenylate-activated protein kinase levels and reduces insulin resistance [44]. Additionally, research has shown that baicalin’s metabolites have the same mechanism owing to the poor oral utilization rate of the drug [59].

Moreover, 3, 5-di-CQA has -glucosidase inhibitory activity, which can reduce intestinal glucose absorption and regulate postprandial hyperglycemia [23].

### 3.2. Targeted Amyloid β-Peptide

An important aspect of T2DM is hyperinsulinemia. Aβ can be degraded by the insulin-degrading enzyme (IDE), which can become less active in response to hyperinsulinemia. The buildup of Aβ in the brain and the emergence of cognitive dysfunction can therefore be facilitated by the decreased activity of IDE [60]. The level of expression of the insulin-degrading enzyme was significantly increased by activating the peroxisome proliferator-activated receptor γ/AMP-activated kinase signaling pathway, the accumulation of Aβ40 and Aβ42 was reduced, and the locational learning and recognition disorders in mice model were alleviated [61].

Aβ is a tiny molecular fragment produced by the amyloid precursor protein (APP)-controlled two-step proteolytic enzymes β-secretase and γ-secretase. Monomers, dimers, polymers, and fibrous polymers like Aβ40 and Aβ42 are all forms it may take [62]. A complicated cascade of inflammatory reactions, microglia activation, and cytokine release accompanies the creation and deposition of SP as a result of increased Aβ production and an increased Aβ42/Aβ40 ratio [63,64,65]. Cognitive impairment, neural abnormalities, and gradual neurogenic damage are the results of these interactions.

Scutellarin decreased the cognitive impairment of APP/PS1 mice in animal models, inhibited the buildup of amyloid, and decreased the levels of solubility Aβ-42 and Aβ-40 in the mouse brain. Scutellarin protected the loss of cognitive function in an in vitro experiment by converting Aβ monomer into elevated amyloid fibrils or fibrils and by lowering the concentration of highly toxic soluble Aβ oligomers [31]. Another investigation further established that scutellarin, when continuously administered, considerably decreased the framework latency and enhanced swimming in APP/PS1 mice. Scutellarin also demonstrated anti-amyloidosis effects, lower levels of proinflammatory cytokines, including tumor necrosis factor (TNF)-α and interleukin (IL)-6, and decreased levels of proinflammatory cytokines, soluble and insoluble Aβ in the brain and bloodstream of mice [36]. Scutellarin showed an anti-Aβ effect in rat models with chronic cerebral hypoperfusion. Four weeks of treatment with 30 mg/kg scutellarin significantly improved spatial cognitive impairment and memory deficits in rats with permanent bilateral common carotid artery occlusion (pBCAO). Additional research revealed that scutellarin inhibited Aβ formation by inhibiting APP expression at beta-site APP-cleaving enzyme 1 (BACE-1) in the hippocampus of pBCAO rats. Furthermore, in the cerebral cortex and hippocampus, scutellarin (30 mg/kg) dramatically reduced the levels of glial fibrillary acidic protein and Iba1, blocking the excitability of glial cells (microglia and astrocytes) in the brain tissue [34].

Previous research has demonstrated that CQA can increase or improve the memory and learning abilities of several AD animal models. For instance, supplementing with 5-CQA dramatically decreased the development of hippocampal Aβ plaques and neuronal death. This neuroprotective effect results from the restoration of aquaporin 4’s perivascular location by 5-CQA and the up-regulation of the target gene low-density lipoprotein receptor-associated protein 1 (Aβ efflux receptor), which promotes Aβ clearance through the perivascular route [24].

### 3.3. Targeted Tau Protein

A versatile protein known as tau is connected to microtubule activity. The transport of nutrients or information molecules between synapses is facilitated by neurons’ stable microtubule structure [66].

By direct and indirect induction, T1DM and T2DM both can phosphorylate tau protein and produce neurofibrillary tangles [67,68]. In direct induction, kinases such as glycogen synthase kinase-3β and protein kinase A are activated, while phosphatase activities such as protein phosphatase 2A (PP2A) are inhibited; in indirect induction, kinase and phosphatase activities are affected by low temperature.

Neurofibrillary tangles and the ultimate onset of AD are caused by hyperphosphorylation of tau, which impairs its capacity to bind to tubulin and encourage microtubule assembly. This undermines microtubule stability and intracellular transport activity [69].

Scutellarin improved horizontal and vertical mobility in the autonomic nerve activity test and decreased exit latency in the Morris water maze (MWM) test in an aluminum chloride plus D-galactose-induced AD mouse. After scutellarin therapy, p-Tau levels considerably decreased, which raises the possibility that this decrease in p-Tau levels may be the cause of the reduced AD-related symptoms. Additionally, acetylcholine and SOD levels in plasma and cerebrospinal fluid have increased, which is indicative of scutellarin’s ability to protect the brain [35].

Protein kinase and dephosphorylated protein phosphatase worked together to phosphorylate the Tau protein. The primary protein kinases that resulted in the phosphorylation of Tau protein were glycogen synthase kinase-3 (GSK-3) and cyclin-dependent protein kinase 5 (CDK-5). Therefore, one of the primary routes that control the phosphorylation of Tau protein may be the PI3K/AKT/GSK-3 signaling pathway. The primary phosphatase of phosphorylated Tau protein is PP2A, and dephosphorylation of this enzyme prevents the development of neurofibrillary tangles (NFT). According to research, quercetin has a neuroprotective impact on glutamic acid-induced HT22 cells and may considerably boost the active expression of PP2A [48]. Additionally, quercetin has been shown to reduce the neuropathology associated with hyperphosphorylation of Tau protein at the Ser 396, Ser 199, Thr 231, and Thr 205 sites by suppressing CDK5 activity [49].

### 3.4. Targeted Neuroinflammation

Since DM is a chronic inflammatory illness, many of its consequences also include inflammatory responses. In DCD, neuroinflammation is significant. Cognitive problems may result from inflammatory substances that cause neuronal injury [70].

T2DM triggers M1-type macrophages to increase the secretion of inflammatory cytokines such as TNF-α and IL-1β, and activate the nucleotide-binding oligomerization domain-like receptor protein 3 (NLRP3) inflammasomes [71]. In the brain, T2DM activates M1-like glial cells, reinforcing the susceptibility to central neuroinflammation. Under normal circumstances, the inflammatory response can serve as a protective mechanism to improve the effects of toxic substances in cerebral neurons, but persistent hyperglycemia can trigger the activation of the nuclear factor kappa-B (NF-κB) pathway and the release of pro-inflammatory factors [72], resulting in the imbalance between pro-inflammatory and anti-inflammatory networks, with the increase of reactive oxygen species and unrestricted formation of inflammatory mediators, thereby affecting mitochondrial function and leading to neuronal damage [73]. In addition, neuroinflammation promotes hyperphosphorylation of Tau and increases IL-1β [74]. Scutellarein can inhibit the release of pro-inflammatory cytokine TNF-α in microglia BV-2 induced by high glucose in vitro [29]. 

NLRP3 is a multi-protein complex, which includes oligomers of NLRP3, apoptosis-associated speck-like protein containing Caspase-recruitment domain (ASC), and caspase-1 [75]. NLRP3 inflammasomes mature and secrete IL-1β and IL-18 under the action of activated β-1 protein [76]. NLRP3 inflammasomes promote diabetes-induced endothelial inflammation and atherosclerosis [77]. Scutellarin reduces the release of inflammatory factors such as IL-1β and IL-18 by inhibiting the production of NLRP3 inflammasomes [78]. In addition, research showed that quercetin could inhibit the expression of inflammation-related proteins regulated by NLRP3, increase the activity of silent information regulation factor 2 homolog-1 (SIRT1), and reduce fasting blood glucose, thereby improving the symptoms of diabetic encephalopathy (DE) [47].

A common cellular enzyme called glyoxalase 1 (Glo-1) is involved in the detoxification of methylglyoxal (MG). Glo-1 can prevent the synthesis of advanced glycation end products (AGEs) and facilitate the elimination of α-carbonyl aldehydes like MG [79,80]. Furthermore, Glo-1 directly prevented the synthesis of AGEs [81,82]. MG is a cytotoxic by-product of glycolysis that causes inflammation, oxidative stress, and the production of AGEs. Apoptosis, ROS generation and the expression of inflammatory cascade enhancers in brain cells are all regulated by the AGEs/RAGE/NF-κB signaling pathway (Figure 2), which can be activated by AGEs [83]. Quercetin improved the symptoms of diabetic encephalopathy, as demonstrated by Zhu’s team, by upregulating Glo-1 to lower AGEs levels, oxidative stress, and inflammation [50].

Baicalin (100 mg/kg) treatment for 14 days in an Aβ1-42-protein-induced mouse model can dramatically improve memory impairments in MWM and probe tests, degrade glial cell activation, and reduce inflammatory factors (IL-6 and TNF-α), revealing that baicalin improves Aβ1-42-protein-related pathological and cognitive disorder through its anti-neuroinflammatory activity [43]. Furthermore, by blocking the activation of the pyrin portion of the NLRP3 family, baicalin greatly reduced the number of activated MG, proinflammatory cytokine levels (IL-1β, IL-18, and iNOS), and neuroinflammation-mediated induction of death. Furthermore, spatial memory impairment in APP/PS1 mice can be alleviated by inhibiting the TLR4/NF-κB signaling cascade and inflammasomes [42].

### 3.5. Targeted Oxidative Stress

Oxidative stress is the principal cause of cell damage and results from an imbalance between the oxidation and anti-oxidation systems in cells and tissues. It is primarily brought on by increased generation and/or impaired clearance of reactive oxygen species (ROS) in the body [84]. 

Diabetes is characterized by hyperglycemia and oxidative stress brought on by fatty acids, and it develops when there is an excessive buildup of ROS due to a deficiency in antioxidant genes [85]. As a result of oxidative stress brought on by excessive ROS generation, which promotes increased metabolism and mitochondrial malfunction, β-cells are less able to secrete insulin [86]. Otherwise, it was discovered that pro-inflammatory cytokines, which serve as signaling molecules and mediators of the inflammatory response, were elevated together with the overexpression of ROS [87]. The most prevalent markers of oxidative stress—superoxide dismutase (SOD), malondialdehyde (MDA), tumor necrosis factor (TNF-α), and interleukin-1β (IL-1β)—can indicate the body’s antioxidant capability. To combat free radicals in the body and use its antioxidant properties, quercetin may lower MDA levels and increase SOD activity [46].

Nuclear factor erythroid 2-related factor 2 (NRf2) is an important antioxidant factor in the body, and its activation will promote the expression of related factors in the antioxidant system and exert antioxidant effects. Scutellarin [32] and baicalin [40] might increase the expression of Nrf2 and its downstream antioxidant factor heme oxygenase-1 (HO-1) and lessen kidney oxidative damage in db/db mice, lowering MDA levels and boosting SOD activity. Scutellarin may lower MDA content in rat hippocampus, boost SOD activity, and dramatically improve learning and memory capacity, according to another research using rats with LPS-induced cognitive impairment as the research subject [37].

### 3.6. Targeted Endoplasmic Reticulum Stress

Endoplasmic Reticulum (ER) stress is a reaction process where cells activate the caspase-12-mediated death pathway in response to calcium homeostasis dysregulation, misfolded and unfolded protein aggregation in the endoplasmic reticulum and the unfolded protein response (UPR). ER stress can cause endoplasmic reticulum molecules to be expressed as chaperones, such as the glucose-regulated proteins GRP78 and GRP94, and as a result have a protective effect. ER stress can also independently cause endogenous cell apoptosis and thus influence the response of stressed cells, such as adaptation, damage or apoptosis.

UPR may activate several signaling proteins to reestablish intracellular equilibrium and is a protective mechanism to preserve ER homeostasis [88]. Inositol-requiring enzyme 1 (IRE1), activating transcription factor 6 (ATF6), and protein kinase RNA like endoplasmic reticulum kinase (PERK) are the three transmembrane proprioception-mediated mechanisms that start UPR.

Studies have shown a strong correlation between ER stress and cognitive impairment, which is a crucial regulatory mechanism for the onset and progression of neurodegenerative disorders as well as a crucial molecular mechanism for diabetes and its consequences [89,90]. Long-term high glucose initiates the NF-κB inflammatory pathway generated by JNK and greatly increases ER stress (Figure 2). Additionally, it was shown that activating the JNK signaling pathway might block insulin signaling and accelerate cognitive loss in the study of the brain injury model [91]. Diabetes also controls the PI3K/Akt signaling pathway and the PERK signaling pathway. As a result, BDNF production was downregulated, cAMP-response element binding protein (CREB) activity was suppressed, and cognitive performance was decreased [92].

A member of the sirtuin family and a deacetylase, SIRT1 is essential for cell differentiation, proliferation, senescence, and death [93]. Three signaling pathways (IRE1α, PERK, and ATF6) that trigger ER stress may be inhibited by upregulating SIRT1 [94,95]. Cognitive impairment and downstream PERK and eIF2α are strongly associated [96]. According to a study [45], quercetin (70 mg/kg) can effectively increase the learning and memory function of db/db mice, lower their inhibited glucose tolerance and insulin resistance, increase the expression of proteins related to nerve and synaptic function (PSD93, PSD95, NGF, and BDNF), and prevent neurodegeneration. Meanwhile, it was shown that quercetin increased the expression of the SIRT1 protein, decreased the expression of proteins connected to the ER signaling pathway (PERK, IRE-1α, and ATF6), and thus decreased the amount of SOD.

### 3.7. Targeted Anti-Apoptosis

Diabetes may increase apoptosis by lowering the expression of many memory-related proteins in hippocampus neurons and decreasing the activity of the PI3k/Akt/CREB signaling pathway. AKT-centered signaling cascade PI3K/Akt is broadly present in all types of cells. Since it participates in cell proliferation, differentiation, and growth control and receives extracellular signals and cell responses, it has drawn significant interest in the treatment of AD [97]. PI3K may be activated by both the protein tyrosine kinase receptor and the G protein-coupled receptor [98], and the activated PI3K can lead to the activation of AKT. An essential downstream target kinase is AKT. Activated AKT may initiate the production of transcription factors such CREB and NF-κB to increase cell survival as well as prevent the production of pro-apoptotic proteins FOXO, Bad, and caspase-9 to directly suppress an anti-apoptotic action [99]. The primary mechanism by which AKT controlled the anti-apoptotic pathway was the phosphorylation and subsequent inactivation of GSK-3β. AKT activity is diminished by inhibiting AKT’s phosphorylation at the Ser 473 and Thr 308 sites, which decreases downstream GSK-3β’s phosphorylation at the Sre 9 site, increases the permeability of the intracellular mitochondrial membrane, produces apoptotic factors, and causes apoptosis [100].

The neurotrophic protein known as a brain-derived neurotrophic factor (BDNF), which was first identified in pig brains in 1982, is primarily expressed in the central nervous system, with the hippocampus and cortex having the highest concentrations. BDNF is crucial for the growth, regeneration, survival, and functional maintenance of neurons. In addition to promoting neuron survival, research has shown that BDNF also controls synaptic plasticity, resulting in presynaptic and postsynaptic long-term potentiation (LTP) [101], an activity that depends on new protein synthesis. Not only are the spatial learning and memory abilities of rats improved in their complex living environment, but hippocampus BDNF mRNA expression is also boosted, and the architecture and quantity of hippocampal neurons are also altered [102]. By controlling the synaptic morphological alterations brought on by LTP and outside inputs, BDNF may govern neuronal information processing and play a significant role in learning and memory.

Tyrosine kinase receptor B (TrkB) is the site of action for BDNF. Through a number of cellular signaling cascades, the binding of BDNF to TrkB activates the intracellular area, increasing TrkB’s autophosphorylation, which in turn activates CREB at the serine site [103]. The P13K pathway is the most significant downstream signaling pathway that can be activated by the combination of BDNF and TrkB [104]. According to Chen et al. [105], stimulation of the PI3K-Akt-mTOR signal transduction pathway exhibited a neuroprotective impact and may be used to stimulate the expression of activity regulated cytoskeleton associated protein (Arc).

After eight weeks of oral treatment of scutellarin (50 mg/kg), mice in the 35 mg/kg streptozotocin (STZ)-induced diabetic mouse model had enhanced learning and memory functions, and IL-β and TNF-α levels were reduced. Procaspase-8, Procaspase-3, Procaspase-9, Cleaved-capacity-8, Cleaved-capacity-9 and Bax were found to have lower protein contents, whereas Bcl-2 was found to have higher protein concentrations. Scutellarin is thought to improve diabetic cognitive impairment by upregulating BDNF/TrkB and its downstream signaling pathway PI3k/Akt/CREB (Figure 2). This is thought to minimize inflammation and hippocampus cell death in T2DM rats [33].

The binding of BDNF to TrkB triggers two significant downstream signaling cascades, the PI3K/Akt and the mitogen-activated protein kinase (MAPK)/extracellular signal-regulated kinase (Erk) pathways. The fundamental function of the MAPK/Erk pathway is to control transcription, which in turn controls cell growth, differentiation, proliferation, apoptosis, and migration [106]. Baicalin has been discovered to control the BDNF-TrkB/PI3K/Akt and MAPK/Erk 1/2 signal axes, therefore reducing astrocyte damage and death [39]. Baicalin significantly decreased the apoptosis that Aβ1-42 caused in SH-SY5Y cells by blocking the Ras-ERK signaling pathway [38].

### 3.8. Targeted Blood–Brain Barrier Injury

The brain microvascular endothelial cells (BMVEC), which are highly specialized and joined by tight junction (TJ) proteins to form the blood-brain barrier (BBB), have a distinct expression of transporters and exhibit minimal phagocytic activity. Pericytes, which surround brain endothelial cells, promote BMVEC activity [107]. The central nervous system (CNS)’s special internal environment is well-protected by the BBB, which also helps to prevent hazardous chemicals from passing through. The loss of TJ proteins (ZO-1, occludin, and claudin-5) and subsequent increase in BBB permeability are both brought on by diabetes, according to recent studies [108]. However, before cognitive impairment and neurodegeneration, BBB integrity was compromised [109]. BBB’s demise may thus be utilized as a sign of DCD before it ever occurs.

Recently, it was discovered that neuroinflammation is strongly associated with BBB injury-induced memory impairment in animal models of DM [110]. An essential mediator of BBB damage is the inflammatory response [108]. TNF-α and IL-1β destroy TJ proteins and control their cellular translocation, therefore suppressing these cytokines and maintaining the BMVEC tight junction [111]. Numerous inflammatory cytokines, such as TNF-α, IL-1β, and IL-6, are upregulated by high glucose levels. Diabetes can increase TNF-α and IL-6 levels, cause the brain’s TJ protein to degrade, aggravate inflammation and leukocyte infiltration, encourage the breakdown of the blood-brain barrier, and ultimately result in cognitive impairment [112]. Diabetes-related BBB failure may also result from interactions between other molecules, such as matrix metalloproteinases. According to research, an increase in matrix metalloproteinase-9 (MMP-9) and tissue inhibitor of matrix metalloproteinase production is associated with an increase in BBB permeability brought on by hyperglycemia [113]. Additionally, diabetes-related oxidative stress has an impact on BBB integrity [114]. The biomarker levels of lipid peroxide and protein peroxide in the brain are markedly elevated after consuming a high-fat meal [115]. To lessen the risk of DCD, it may be possible to stop diabetes from damaging the BBB (Figure 2).

Scutellarin prevents blood–brain barrier damage caused by cerebral ischemia, by inhibiting iNOS expression and decreasing MMP-9 transcription and synthesis. 3,5-di-CQA protects the blood-brain barrier from damage induced by cerebral ischemia by lowering iNOS expression, suppressing ROS/RNS generation, MMP-9 transcription and synthesis, and reducing MMP-9 degradation mechanisms on claudin-5, occludin, and ZO-1 [28]. By preserving the integrity of TJs, scutellarein may lessen the production of inflammatory markers, such as TNF-α, in microglia produced by high hyperglycemia and thus reverse BBB damage [29].

### 3.9. Targeted Gut Dysbacteriosis

Previous research has demonstrated that scutellarin has limited oral bioavailability, owing to its low solubility in gastrointestinal fluids, poor membrane permeability, first-pass metabolism in the gastrointestinal tract, and transporter efflux. Scutellarin is absorbed mostly by passive diffusion in the gastrointestinal system, and absorption is linear with concentrations ranging from 50 to 400 μg/mL. When pH is 6.0-7.4, absorption is unaffected by pH, and the kidney has the greatest content distribution in the body, followed by the heart, liver, and brain [116]. Before absorption, scutellarin is degraded by microbial glucuronidase into scutellarein [117]. Scutellarein was shown to be more readily absorbed than scutellarin following oral administration in prior research [118]. As a result, increasing research has been conducted in recent years to investigate the association between gut flora and diabetic cognitive impairment [119].

Intestinal flora imbalance may cause a number of disorders, including obesity, diabetes, and Parkinson’s disease (PD), and it is implicated in the cognitive impairment process of Alzheimer’s disease (AD) [120]. Restoring gut flora equilibrium will relieve cognitive impairment and neuropsychiatric symptoms [121]. The brain-gut axis affects cognitive performance through intestinal flora and its metabolites, according to research [122,123]. The interactive communication between the brain and the gut is completed by signaling pathways that are mutually overlapping and complimentary.

The microbiota-gut-brain axis and metabolites, particularly short-chain fatty acids (SFCAs) generated by intestinal bacterial fermentation, are thought to play a significant role in the microbiota-gut-brain connection [124,125]. It should be noted that two traditional SFCAs, butyrate and acetate, were discovered in T2DM and T1DM, respectively [126,127]. Acetate-producing bacteria consumption causes long-term acetate deficit and decreases synaptic proteins in the hippocampus, causing cognitive impairment in T1DM mice [127]. The function of SFCA in T2DM-induced cognitive deterioration, on the other hand, is unknown.

Microglia, the brain’s primary immune cells, can interact with gut bacteria or metabolites [128]. A recent study found that intestinal bacteria influenced microglia development and function via microbial-derived SCFAs [129]. Single-chain fatty acids have been shown to block cAMP-PKA-CREB-HDAC signal transduction, perhaps aiding in the prevention of Alzheimer’s disease [130].

After feeding APP/PS1 mice scutellarin (100 mg/kg) for two months, Zhang’s team [27] discovered that the mice’s learning memory and object recognition abilities were significantly improved. The abundance of the microflora in the feces was dominated by the Prevotellaceae, Bacteroidales, and Staphylococcaceae families after drug treatment. A Spearman correlation study revealed a negative association between the brain’s level of IL-1β and changes in the composition of the gut flora and the concentration of Aβ oligomers. Through controlling the intestinal flora, reversing cAMP-PKA-CREB-HDAC3 signaling in microglia to lower the number of Aβ oligomers and down-regulating the pro-inflammatory cytokine IL-1β, scutellarin was shown to enhance cognition in AD rats (Figure 2).

## 4. Conclusions

Current research suggests that many factors contribute to the pathogenesis of DCD, including insulin resistance, oxidative stress, neuroinflammation, endoplasmic reticulum stress, blood–brain barrier damage, apoptosis, an imbalance in the intestinal flora and the buildup of amyloid β-peptide and tau proteins. Diabetes patients may receive similar treatment to other non-diabetic patients once the diagnosis of cognitive dysfunction has been made. Cognitive enhancers such as cholinesterase inhibitors (e.g., donepezil, rivastigmine, or galanthamine) or N-methyl-D-aspartic acid antagonists may be used in such therapies, but their clinical efficacy is lower. Furthermore, delaying the progression of DCD is largely dependent on blood glucose control [131].

As a result, the future research path of DCD medications will be of a multi-target design. TCM has the synergistic impact of numerous components, multiple routes and various targets in the therapy of DCD and it has significant promise. Many Chinese medicines, including EBHM, show considerable benefits in treating cognitive impairment and further study is needed. We discovered that the main ingredients of EBHM, such as scutellarin, baicalin, quercetin and CQA, have strong pharmacological effects and research possibilities in the treatment of DCD. The processes include lowering insulin resistance and A toxicity, suppressing tau phosphorylation, resisting oxidative stress and neuroinflammation, providing anti-apoptotic action, controlling gut flora, alleviating blood–brain barrier damage and anti-endoplasmic reticulum stress response. At the same time, we discovered in our research that AMPK and CREB agonists may be highly effective medications for treating DCD symptoms (Figure 2). AMPK has been shown in studies to act as a link between DM and AD [132]. AMPK agonists are currently mostly tiny compounds with little selectivity. When AMPK is activated, additional kinases, such as glycogen phosphorylase, may be activated [133]. More research should be conducted in the future to concentrate on the direct recognition of agonists working on AMPK and CREB targets.

As a therapeutic medication for DCD, EBHM has a long way to go. To begin with, EBHM anti-DCD investigations have been focused on animal models and have not been used in controlled clinical trials in DCD patients. Second, quercetin, one of the flavonoids in EBHM, has a favorable therapeutic impact on DCD, but no EBHM preparations are presently available on the market [17]. According to research, the vacuum reflux extraction technique extracted 0.0684 mg/kg [134] of quercetin from EBHM. Furthermore, the synergistic effects of multiple active components of EBHM on DCD are still unknown. Finally, despite much research on animal models of DCD, the doses of EBHM and its active components differ. According to studies [135], apigenin-O-7-β-D-glucuronic acid, 1, 5-di-o-caffeoylquinic acid, and 3, 4-di-o-caffeoylquinic acid are EBHM components that preserve brain neuronal function. However, the active components of EBHM are mostly focused on scutellarin, and no study has been conducted on other components that may have therapeutic benefits on DCD.

Therefore, it is essential to carry out network pharmacology on EBHM in the future to offer suggestions for the creation of novel EBHM preparations and to create the groundwork for upcoming clinical research on the use of EBHM for the treatment of DCD. At the same time, the primary innovation in future research may be the creation of a drug delivery system with straightforward preparation technology, high drug loading, good oral absorption and better bioavailability.

## Figures and Tables

**Figure 1 pharmaceuticals-16-00050-f001:**
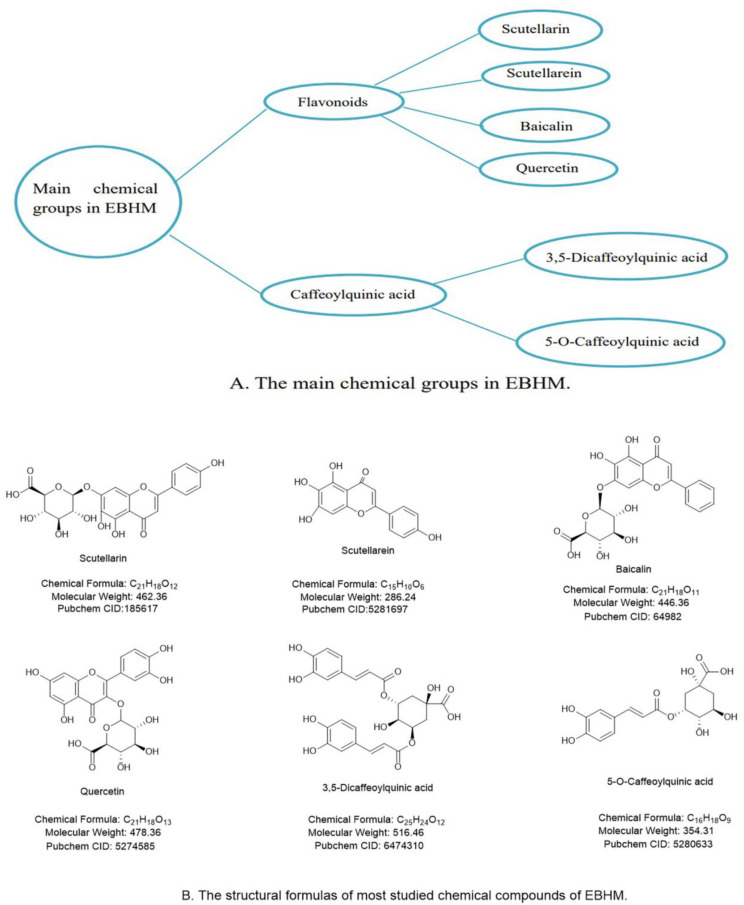
Active ingredients in EBHM with anti-DCD characteristics. (**A**) The main chemical groups in EBHM; (**B**) The structural formulas of most studied chemical compounds of EBHM.

**Figure 2 pharmaceuticals-16-00050-f002:**
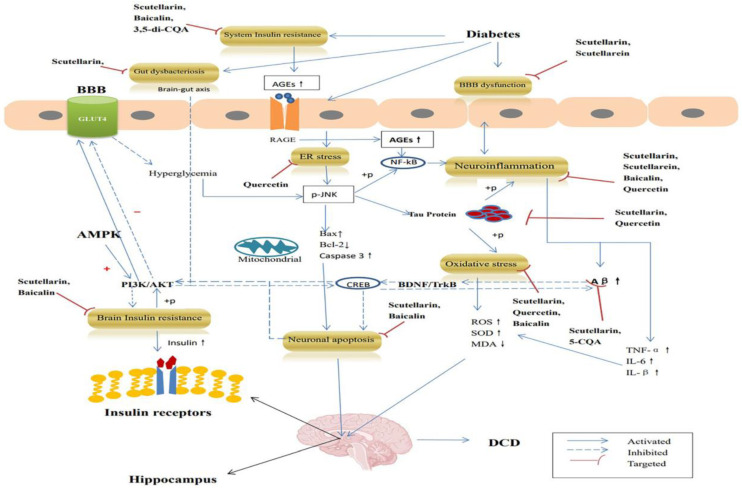
The anti-DCD mechanism of EBHM’s active component.

**Table 1 pharmaceuticals-16-00050-t001:** The pharmacology and probable mechanisms of EBHM’s compounds and metabolism for treating DCD.

Active Ingredients	Model	Administration	Pharmacological Actions	Test Index	Possible Mechanism	References
Scutellarin	APP/PS1 mice	20/100 mg/kg, p.o.	Regulates intestinal microbiota, decreases Aβ oligomer and downregulates IL-1β by reversing cAMP-PKA-CREB-HDAC3 in microglia.	OFT, NORT	Targeting the composition of intestinal microbiota, Aβ, neuroinflammation	[27]
Scutellarin	MCAO rats	0.5 mg/kg, i.v.	Inhibits MMP-9 transcription	—	Targeting BBB dysfunction	[28]
Scutellarein	BV2 cells	20/50 μM	Reduces TNF-α expression and maintains the integrity of TJs	—	Targeting BBB dysfunction, neuroinflammation	[29]
Scutellarin	C57BL/6 mice	50 mg/kg, p.o.	Activates AMPKα, promotes glucose uptake and increases SOD levels	FBG, INS, HOMA-IR, HOMA-IS, HOMA-β, QUICKI, IPGTT, IPITT	Targeting insulin-resistance	[30]
Scutellarin	APP/PS1 mice	50 mg/kg, i.v.	Lowers cortical levels of soluble human Aβ42 and Aβ40	EPM, MWM	Targeting Aβ	[31]
Scutellarin	db/db mice	25/50/100 mg/kg, i.g.	Increases SOD activity inhibits MDA production and reduces IL-1β expression	OGTT	Targeting oxidative stress, neuroinflammation	[32]
Scutellarin	STZ (35 mg/kg)-induced diabetic rats	50 mg/kg, i.g.	Increases SOD levels, reduces Bcl-2 levels	MWM, OFT, FBG, INS, HOMA-IR, ISI	Targeting anti-apoptosis, oxidative stress, neuroinflammation	[33]
Scutellarin	pBCAO rats	30 mg/kg, p.o.	Reduces the production of Aβ by preventing the expression of APP and BACE-1 and prevents glial cell activation	MWM	Targeting Aβ, neuroinflammation	[34]
Scutellarin	Balb/c male mice(D-gal, AlCl3)	20 mg/kg, p.o.	Increases acetylcholine and SOD levels while decreasing p-tau and Aβ42 levels	MWM	Targeting t126au protein, Aβ, oxidative stress	[35]
Scutellarin	APP/PS1 mice	50 mg/kg, p.o.	Lowers the expression of pro-inflammatory cytokines and decreases Aβ expression in the brain and plasma	MWM	Targeting Aβ, neuroinflammation	[36]
Scutellarin	LPS (500 μg/kg)-induced cognitive impairment mice	5/25/50 mg/kg, i.g.	Increases SOD activity decreases MDA activity and decreases the level of proinflammatory cytokines	Y-maze, novel object recognition, passive avoidance test	Targeting oxidative stress, neuroinflammation	[37]
Baicalin	SH-SY5Y cells	5/10/20 µM	Improves SH-SY5Y cells activity	—	Targeting anti-apoptosis	[38]
Baicalin	Neuron–Astrocyte Cocultures	34.38 μg/ml	Increases SOD activity inhibits reactive astrocytes production and reduces IL-1β expression	—	Targeting anti-apoptosis, oxidative stress, neuroinflammation	[39]
Baicalin	db/db mice	400 mg/kg, i.g.	Increases SOD activity decreases MDA activity and decreases the level of proinflammatory cytokines	FBG, OGTT, ITT, HOMA-IR	Targeting oxidative stress, neuroinflammation	[40]
Baicalin	Male C57BL/6 mice	200/400 mg/kg, i.g.	Upregulates Insulin receptor expression, and reduces TNF-α expression by modulating macrophage differentiation in the adipose tissue and liver.	OGTT, OFTT, HOMA-IR	Targeting insulin resistance, neuroinflammation	[41]
Baicalin	APP/PS1 mice	100 mg/kg, i.p.	Reduces the quantity of pro-inflammatory cytokines and the amount of activated microglia	MWM, probe test	Targeting neuroinflammation	[42]
Baicalin	Male ICR mice (Aβ-injected hippocampus)	100 mg/kg, i.g.	Reduces inflammatory factor (IL-6, TNF-α) expression and glial cell activation	MWM, probe test	Targeting neuroinflammation	[43]
Baicalin	STZ (45 mg/kg)-induced diabetic rats	80 mg/kg, i.p.	Activates AMPKα, promotes glucose uptake, and inhibits MDA production	—	Targeting insulin-resistance	[44]
Quercetin	db/db mice	70 mg/kg, i.g.	Upregulates SIRT1 protein expression and inhibits the expression of ER signaling pathway-related proteins	OGTT, ITT, MWM	Targeting ER stress	[45]
Quercetin	STZ(60 mg/kg)-induced diabetic rats	50 mg/kg, i.g.	Increases SOD activity decreases MDA activity and decreases the level of proinflammatory cytokines	FBG	Targeting oxidative stress, neuroinflammation	[46]
Quercetin	db/db mice	70 mg/kg, i.g.	Decreases NLRP3 inflammation-related proteins and inflammatory factor (IL-1β and IL-18) expressions	MWM, NORT, OGTT, IRT	Targeting oxidative stress, neuroinflammation	[47]
Quercetin	glutamate-exposed HT22 cells	5/9 µM	Decreases p-tau	—	Targeting tau protein	[48]
Quercetin	okadaic acid-exposed HT22 cells	1/3/5 µM	Decreases p-tau	—	Targeting tau protein	[49]
Quercetin	STZ (60 mg/kg)-induced diabetic rats	90 mg/kg, i.g.	Increases AGEs levels by inhibiting Glo-1 expression, reduces IL-β, and TNF-α expression and increases SOD levels	FBG	Targeting oxidative stress, neuroinflammation	[50]
5-CQA	APP/PS2 transgenic mice	0.8% (*w*/*w*)	Modulates Aβ and neuronal loss	Y-maze, novel object recognition	Targeting Aβ	[24]
3,5-di-CQA	STZ (40 mg/kg)-induced diabetic rats	100/200 mg/kg, i.p.	Inhibits α -glucosidase activity	OGTT	Targeting insulin resistance	[23]

## Data Availability

The datasets used during the study are available from the correspond ing author upon reasonable request.

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
