# Peer review of "Advances in Anti-Diabetic Cognitive Dysfunction Effect of Erigeron Breviscapus (Vaniot) Hand-Mazz"

_pharmaceuticals, 2022, doi:10.3390/ph16010050_

Round 1
Reviewer 1 Report
Major concern
Cognitive impairment results from chronic disease outcomes from the early onset of diabetes in life. Recent systematic review and meta-analysis (10.1002/jmv.27996) documented that the new-onset of type 1 diabetes has increased worldwide during the pandemic period. This issue should be mentioned in the introduction to explain the current review's importance.
Apart from EBHM, there are several other herbs like Urtica dioica that are capable to improve cognitive function, neuroinflammation, and brain antioxidants in diabetic conditions. This issue should be mentioned in the introduction.
Minor comments
Please revise the following parts in the abstract: ‘Even though their etiology is unknown, the mechanisms that produce DCD do not exist separately. It's hard to tell if DCD is a direct result of diabetes or a mix of its effects because its processes overlap with broad non-specific neurodegenerative disease’
Reviewer 2 Report
Review for Pharmaceuticals (ISSN 1424-8247) manuscript entitled "Advances in anti-diabetic cognitive dysfunction effect of Erigeron breviscapus (Vant.) Hand-Mazz".
In this review the authors have discussed a very interesting and actual question related to possible treatment of diabetic cognitive dysfunction with a traditional Chinese medicine herb Erigeron breviscapus (Vant.) Hand-Mazz (EBHM). The problem is of great importance as diabetes is becoming more and more common and is a global challenge of modern health care all over the world. The cognitive decline is, unfortunately, a common complication of diabetes. In addition, at the moment there is no effective treatment for this pathology. In the work, the authors describe in detail the possible mechanisms of action of the EBHM active compounds at the level of the involved molecular-genetic pathways and biological processes.
The manuscript is an interesting and well-done review, it was carried out at a high scientific level. This work will attract the interest of relevant specialists. The manuscript could be recommended for publication after some minor improvements.
Please, improved/corrected the following:
1. In line 38 it seems that there is needed a comma, not a dot, after “function [3]”
2. It would be useful to add some information in the section “2. Active ingredients and pharmacological effects of EBHM” about how is the EBHM used, for example as a tea, or as alcohol extract, or other.
3. The authors have to check carefully all the references, that they are correct and go in the right order. Please, pay a special attention for references in the Table 1.
4. In line 397 it is better to write the whole name of Arc like “activity regulated cytoskeleton associated protein (Arc)”.
5. At Figure 1 it is better to make subtitles, something like “A. The main chemical groups in EBHM. B. The structural formulas of most studied chemical compounds of EBHM.”
6. It seems that it is better to refer to the Figure 2 not only in the section 4, but also in other places of the text, where the relevant information is states. For example, the oxidative stress is mentioned at the Figure 2 and it is discussed in the section 3.5, and so on. So, I suggest that the Figure 2 could be moved to the middle part of the manuscript and referred multiple times in the text.

Round 2
Reviewer 1 Report
Congratulation to the authors for the work done in the revised version.